# Non-canonical RNA-directed DNA methylation participates in maternal and environmental control of seed dormancy

Mayumi Iwasaki[1], Lena Hyvärinen[1], Urszula Piskurewicz[1], Luis Lopez-Molina[1,2]*

[1]Department of Plant Biology, University of Geneva, Geneva, Switzerland; [2]Institute for Genetics and Genomics in Geneva (iGE3), University of Geneva, Geneva, Switzerland

**Abstract** Seed dormancy is an adaptive trait preventing premature germination out of season. In a previous report (Piskurewicz et al., 2016) we showed that dormancy levels are maternally inherited through the preferential maternal allele expression in the seed endosperm of *ALLANTOINASE* (*ALN*), a negative regulator of dormancy. Here we show that suppression of *ALN* paternal allele expression is imposed by non-canonical RNA-directed DNA methylation (RdDM) of the paternal *ALN* allele promoter. Dormancy levels are further enhanced by cold during seed development. We show that DNA methylation of the *ALN* promoter is stimulated by cold in a tissue-specific manner through non-canonical RdDM, involving RDR6 and AGO6. This leads to suppression of *ALN* expression and further promotion of seed dormancy. Our results suggest that tissue-specific and cold-induced RdDM is superimposed to parental allele imprints to deposit in the seed progeny a transient memory of environmental conditions experienced by the mother plant.
DOI: https://doi.org/10.7554/eLife.37434.001

## Introduction

Newly produced seeds exhibit dormancy, a trait whereby germination is blocked upon imbibition under otherwise favorable germination conditions. Dormancy prevents germination out of season and facilitates final dispersal of the plant embryo. The Arabidopsis seed consists of an embryo surrounded by a single cell layer of endosperm and an external seed coat. The endosperm is essential to repress dormant seed germination by releasing the phytohormone abscisic acid (ABA), which blocks embryonic growth (*Lee et al., 2010*; *Chahtane et al., 2017*). Dry seeds gradually lose dormancy over a period of time, referred as 'after-ripening', thus acquiring the capacity to germinate. After-ripening time is therefore a measure of dormancy levels, which vary across Arabidopsis accessions.

In a previous report, we observed that hybrid seeds arising from reciprocal crosses between high and low dormancy accessions could inherit dormancy levels more akin to the maternal ecotype. We identified a set of genes exhibiting preferential maternal allele expression (genomic imprinting) in the endosperm of dormant seeds, which included seed germination regulators (*Piskurewicz et al., 2016*). We showed that maternal inheritance of dormancy levels is implemented through the preferential maternal allele endosperm expression of *ALLANTOINASE* (*ALN*), a negative regulator of seed dormancy. We also reported that cold temperatures during seed development, which markedly increases dormancy levels in Arabidopsis seeds (*Donohue et al., 2008*; *Penfield and MacGregor, 2017*), suppressed the expression of numerous maternally expressed genes (MEGs), including *ALN*, upon seed imbibition (*Piskurewicz et al., 2016*).

Genomic imprinting requires 'imprints' distinguishing parental alleles. In Arabidopsis, DNA methylation serves as a primary imprint for numerous MEGs (*Rodrigues and Zilberman, 2015*). The DNA

*For correspondence:
Luis.lopezmolina@unige.ch

Competing interests: The authors declare that no competing interests exist.

demethylase *DEMETER* (*DME*) removes methylated cytosines at specific loci of the central cell maternal genome, resulting, after fertilization, in different DNA methylation levels between maternal and paternal genomes (*Choi et al., 2002*; *Choi et al., 2004*; *Gehring et al., 2006*; *Morales-Ruiz et al., 2006*). DNA METHYLTRANSFERASE 1 (MET1) is also required for imprinting by maintaining paternal allele CG methylation during DNA replication (*Jullien, 2006*). In addition to this mechanism, canonical RNA-directed DNA methylation (RdDM) can contribute to paternal allele silencing in some imprinted genes (*Vu et al., 2013*). Canonical RdDM involves RNA polymerase IV and V (Pol IV, Pol V). Pol IV-dependent RNA transcripts are transcribed by RNA-dependent RNA polymerase 2 (RDR2) to produce double-stranded RNAs (dsRNAs). dsRNAs are cleaved by DICER-LIKE 3 (DCL3) into 24-nt small interfering RNAs (siRNAs, which are then loaded into ARGONAUTE 4 (AGO4). The transcripts emerging from Pol V are recognized by siRNAs bound to AGO4 through sequence complementarity. At this stage, DOMAIN REARRANGED METHYLTRANSFERASE 2 (DRM2) is recruited and methylates cytosine in all sequence contexts (CG, CHG, and CHH; H stands for A, C, or T) (*Castel and Martienssen, 2013*; *Matzke et al., 2015*). Non-canonical RdDM pathways, that is pathways not involving Pol IV-RDR2-DCL3, have recently been identified but are less well characterized (*Cuerda-Gil and Slotkin, 2016*).

Our previous report left unanswered the question of the nature of the mechanism leading to genomic imprinting in the endosperm of mature seeds. Furthermore, the mechanisms allowing mature seeds to 'remember' past cold temperatures in order to further regulate imprinted gene expression was not investigated. Here we addressed these questions by focusing on the case of *ALN*.

## Results

### *ALN* imprinting is maintained in *met1* mutants and lost in *drm1drm2* mutants

To investigate whether *ALN* paternal allele suppression requires DNA methylation, we pollinated WT Cvi plants with pollen from methyltransferase mutants, *met1* (Col ecotype) and *drm1drm2*, hereafter referred as *drm* mutants (Ws), lacking functional DRM1 and DRM2.

Preferential maternal *ALN* allele expression was retained in WT x *met1* hybrid seed endosperm but not in that of WT x *drm* (*Figure 1A*). We also pollinated WT Cvi plants with *nrpd1* (Col-0) pollen, deficient in the largest subunit of Pol IV. Interestingly, WT x *nrpd1* hybrid seed endosperm retained preferential maternal *ALN* allele expression (*Figure 1A*).

Altogether, these results show that *ALN* imprinting requires non-canonical RdDM for paternal allele silencing rather than MET1 or canonical RdDM as previously reported (*Gehring, 2013*; *Vu et al., 2013*; *Rodrigues and Zilberman, 2015*). We therefore hypothesized that *ALN* imprinting necessitates DNA methylation marks deposited by non-canonical RdDM. Thus, we explored whether genomic DNA neighboring *ALN* could indeed be targeted by non-canonical RdDM.

### *ALN* upstream sequences are targeted by non-canonical RdDM

The 5' and 3' flanking regions of *ALN* contain transposable elements (TEs), which are often associated with DNA methylation (*Figure 1B*). We analyzed DNA methylation levels in a region of about 1.3 kbp spanning −1500 to −200 bp relative to *ALN*'s transcription start site (TSS), and in a region of about 1.7 kbp spanning −200 to +1500 bp relative to *ALN*'s transcription termination site (TTS) (*Figure 1B*). We identified a heavily methylated region of about one kbp in the 3' flanking region and a moderately methylated region of about 300 bp located about 1.0 kbp upstream of ALN's TSS (*Figure 1B*). Downstream of *ALN*'s TTS the levels of CHH methylation, which can be mediated by RdDM, were markedly reduced in *drm* mutants and *nrpd1* mutants (*Figure 1B*). In contrast, in the methylated region upstream of *ALN*'s TSS, CHH methylation levels in *nrpd1* were similar to WT levels unlike those in *drm*, which were markedly lower (*Figure 1B*). We also analyzed DNA methylation upstream of *ALN*'s TSS in *nrpe1*, deficient in the largest subunit of Pol V, and *met1* mutants. Overall CHH methylation levels in *met1* were similar to WT levels whereas they were markedly lower in *nrpe1* mutants (*Figure 1C*, and *Figure 1—figure supplement 1*).

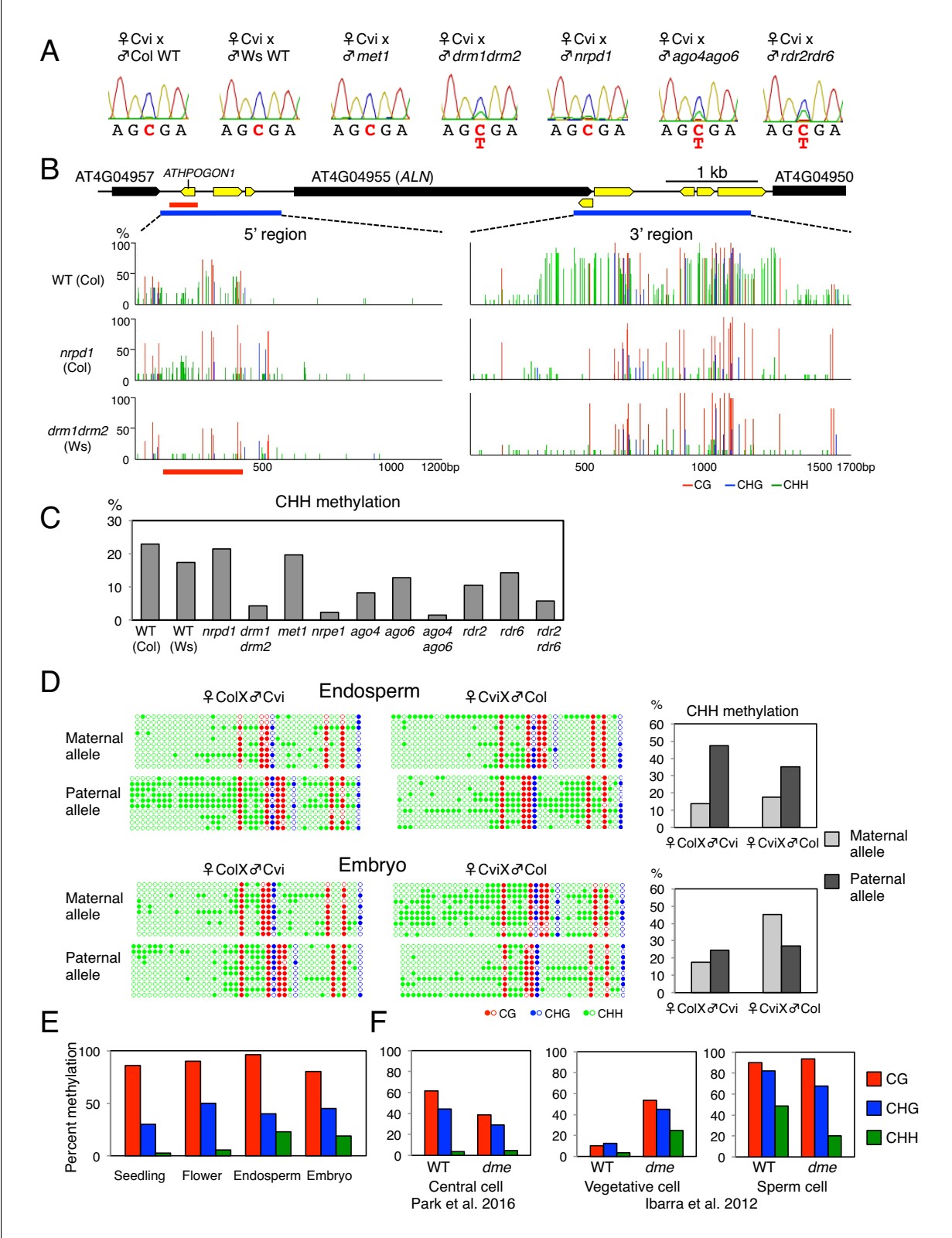

**Figure 1.** *ALN* imprinting necessitates non-canonical RdDM. (**A**) Sanger sequencing chromatograms at SNPs of *ALN*. WT plants (Cvi) were pollinated with WT (Col and Ws) and mutant pollen as indicated. RNA was extracted from endosperm of F1 seeds and subjected to RT-PCR followed by Sanger sequencing. Nucleotides at SNP sites are highlighted in red: 'C' originates from Cvi and 'T' originates from Ws and Col. Representative results are shown from experiments repeated at least 5 times for each genotype. (**B**) DNA methylation on the *ALN* 5' and 3' flanking regions in endosperm of

*Figure 1 continued*

mature seed. Black box arrows and yellow box arrows show genes and TEs respectively. Blue lines show regions where DNA methylation was studied. Red line (300 bp) shows *ALN*'s highly methylated 5' upstream region (referred as 'POGO region'). All remaining methylation data shown in this figure only correspond to the 300 bp POGO region. Red, blue and green vertical lines represent CG, CHG, and CHH methylation levels, respectively. (C) Percentages of CHH methylation levels corresponding to the POGO region in different RdDM mutants (Materials and methods). *drm1drm2* is in Ws background; *ago4ago6* is generated after crossing *ago4* with *ago6* in Ler and C24 background, respectively; all other mutants are in Col-0 background. Percentages of DNA methylation at CG and CHG sites are shown in *Figure 1—figure supplement 1*. (D) DNA methylation levels in maternal and paternal alleles. DNA extracted from endosperm and embryo of F1 seeds obtained after reciprocally crossing Cvi and Col WT plants was analyzed by sodium bisulfite sequencing. SNPs were used to distinguish maternal and paternal alleles. Filled and open circles represent methylated and unmethylated cytosines, respectively. (E) Percentage of DNA methylation in the POGO region in different tissues. (F) Percentage of DNA methylation in the POGO region in female and male gametes. The data were extracted from published whole genomic DNA methylation data of female (*Park et al., 2016*) and male (*Ibarra et al., 2012*) gametes.

DOI: https://doi.org/10.7554/eLife.37434.002
The following source data and figure supplement are available for figure 1:

**Source data 1.** *Figure 1B* and *2B*: DNA methylation in the 5' and 3' flanking regions of *ALN* in the endosperm of mature seeds.
DOI: https://doi.org/10.7554/eLife.37434.004
**Source data 2.** *Figure 1C*: Percentages of DNA methylation levels in the POGO region of different RdDM mutants.
DOI: https://doi.org/10.7554/eLife.37434.005
**Figure supplement 1.** DNA methylation in the POGO region.
DOI: https://doi.org/10.7554/eLife.37434.003

Interestingly, the methylated region upstream of *ALN*'s TSS contains a TE belonging to the *Ath-POGON1* family previously reported to be a target of non-canonical RdDM mediated by RDR6 and AGO6 (referred as RDR6-RdDM) (*Nuthikattu et al., 2013*; *McCue et al., 2015*; *Cuerda-Gil and Slotkin, 2016*). This methylated region is hereafter referred to as 'POGO region'. Indeed, in *ago6* and *rdr6* mutants the POGO region CHH methylation levels were lower than those of WT, although not as low as those found in *drm* or *nrpe1* mutants (*Figure 1B*, and *Figure 1—figure supplement 1*). CHH methylation levels were also decreased in *rdr2* and *ago4* mutants and further decreased in *rdr2rdr6* and *ago4ago6* double mutants (*Figure 1C*, and *Figure 1—figure supplement 1*).

Altogether, these results show that the POGO region is methylated through a form of RdDM not involving Pol IV, that is a non-canonical form, and involving RDR6 and AGO6 in addition to RDR2 and AGO4.

To further investigate whether CHH methylation in the POGO region affects *ALN* imprinting, we pollinated WT plants with pollen from *rdr2rdr6* and *ago4ago6* double mutants. Preferential maternal *ALN* allele expression was lost both in WT x *rdr2rdr6* and WT x *ago4ago6* hybrid seeds (*Figure 1A*). These results consolidate the notion that non-canonical RdDM is required for *ALN* imprinting. This could involve CHH methylation of the POGO region upstream of *ALN*'s TSS.

We previously showed that seed dormancy levels can be regulated by maternal *ALN* allele expression. Indeed, we showed that *aln* x WT hybrid F1 seeds obtained after pollinating *aln* plants with WT pollen have higher dormancy than WT seeds (*Piskurewicz et al., 2016*). To further assess the role of RdDM in maternal regulation of seed dormancy, we pollinated *aln* mutant plants with WT and *nrpe1* pollen. Dormancy levels in *aln* x *nrpe1* hybrid F1 seeds were lower relative to *aln* x WT hybrid seeds, further indicating that maternal regulation of seed dormancy requires suppression of *ALN* paternal allele by RdDM (*Figure 1—figure supplement 1C*).

## The paternal POGO region has higher CHH methylation

To assess whether CHH methylation of the POGO region could be involved in suppressing *ALN*'s paternal allele, we measured allele-specific methylation in the POGO region of hybrid seeds obtained after reciprocally crossing Col and Cvi plants. In embryos, CG methylation levels were similar between parental alleles whereas CHH methylation levels were higher in Cvi alleles (*Figure 1D* and *Figure 1—figure supplement 1*). This bias likely reflects accession differences in DNA methylation levels in the POGO region.

In the endosperm, CG DNA methylation levels were also similar between parental alleles. However, CHH methylation levels were markedly higher in the paternal allele relative to the maternal alleles (*Figure 1D* and *Figure 1—figure supplement 1*).

Altogether these results support the notion that preferential maternal *ALN* allele expression involves *ALN* paternal allele suppression mediated by methylation of the POGO region through non-canonical RdDM.

*ALN* expression is not silenced in seedlings and flowers (*Schmid et al., 2005*; *Watanabe et al., 2014*). We observed that seedlings and flowers had similar CG methylation levels in the POGO region relative to the mature endosperm or embryo. In contrast, they had markedly lower CHH methylation levels (*Figure 1E*).

This further suggests that methylation of CHH sites in the POGO region, mediated by non-canonical RdDM, rather than CG sites, which are present at low density in the POGO region, are involved in suppressing *ALN* paternal allele expression.

### Preferential paternal POGO region methylation likely originates in the male germ line

Available genomic methylation data revealed that CHH methylation levels in the POGO region are low (about 4%) in the central cell (*Figure 1F* and *Figure 1—figure supplement 1*) (*Calarco et al., 2012*; *Ibarra et al., 2012*; *Hsieh et al., 2016*; *Park et al., 2016*). DME is active in the central cell where it establishes imprinted gene expression in numerous imprinted genes after removing DNA methylation (*Choi et al., 2002*; *Park et al., 2016*). In *dme* mutants CHH methylation levels in the central cell remain low and similar to those in WT plants (*Figure 1F*). Thus, in the case of *ALN*, higher paternal CHH methylation in the POGO region does not result from DME-dependent CHH methylation removal in the central cell.

In the case of sperm cells, different POGO region methylation levels are reported by three different groups but they are systematically higher (48%, 34% and 9%) than those reported in the central cell (*Figure 1F* and *Figure 1—figure supplement 1*) (*Calarco et al., 2012*; *Ibarra et al., 2012*; *Hsieh et al., 2016*). In addition to the central cell, DME is also active in the vegetative cell of pollen (*Schoft et al., 2011*; *Ibarra et al., 2012*). In WT, CHH methylation levels are low in the vegetative cell (about 3.5%), but they increase markedly (about 25%) in *dme* mutants, suggesting that DME promotes low CHH methylation in the vegetative cell (*Figure 1F*). In *dme* mutants, CHH methylation levels in sperm cells are reduced but they remain higher (>20%) relative to those in the central cell (*Figure 1F*) (*Ibarra et al., 2012*).

Altogether, these data suggest that preferential paternal *ALN* allele POGO region CHH methylation is a phenomenon specific to the male germ cell lineage, which is further maintained in the endosperm after fertilization. In turn, this leads to paternal allele suppression of *ALN* expression in the endosperm upon imbibition of mature dormant seeds.

### Cold temperatures during seed development increase methylation of the POGO region through RDR6-RdDM

Seed development under cold temperatures leads to increased seed dormancy levels, which is associated with low *ALN* expression upon seed imbibition (*Piskurewicz et al., 2016*; *Penfield and MacGregor, 2017*). Interestingly, *ALN* imprinting was lost in seeds produced at 10 ˚C suggesting that cold perturbs the DNA methylation imprints associated with *ALN* (*Figure 2A*). In seeds produced at 10 ˚C, we observed that the length of the methylated POGO region expanded by 200 bp and CHH methylation levels markedly increased while the DNA methylation levels downstream of *ALN*'s TTS were similar to those observed in seeds produced at 22 ˚C (*Figure 2B*).

Cold-induced CHH methylation was observed in *nrpd1* but not in *drm* or *nrpe1* mutants (*Figure 2C* and *Figure 2—figure supplement 1*). This strengthens the notion that the POGO region is a target of non-canonical RdDM. Indeed, cold-induced CHH methylation was dependent on RDR6 and AGO6 but not on RDR2 or AGO4 (*Figure 2C* and *Figure 2—figure supplement 1*). These results therefore show that non-canonical RDR6-RdDM is required for cold-induced CHH methylation of the POGO region.

### Cold-dependent and tissue-specific stimulation of *AGO6* expression likely drives cold-induced methylation of the POGO region in seeds

Increased CHH methylation in the POGO region was observed both in the endosperm and embryo from seeds produced at 10 ˚C, but not in the flowers from plants cultivated at 10 ˚C (*Figure 2D*).

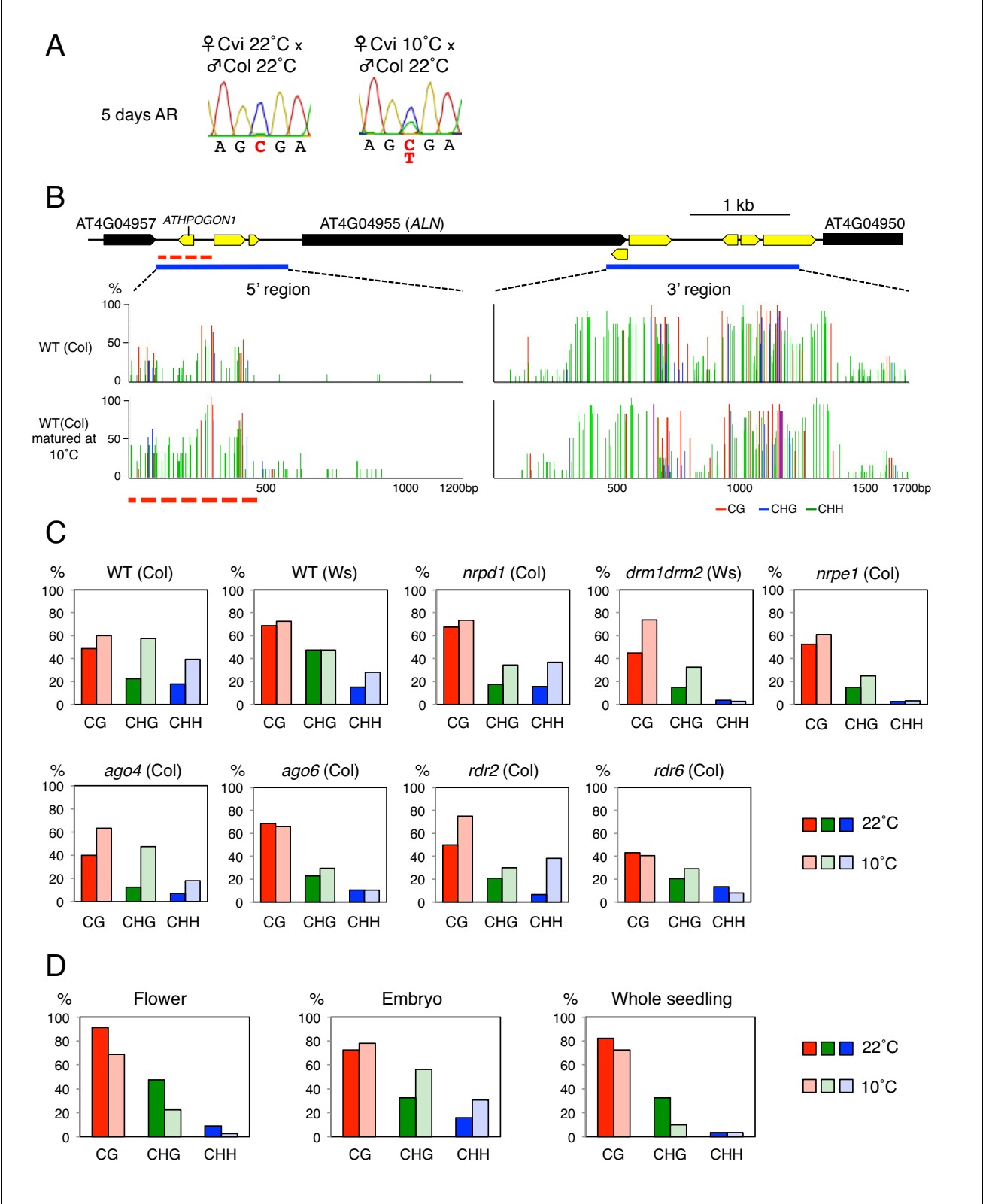

**Figure 2.** Cold temperatures increase methylation levels and expand the length of the *ATHPOGON1*-containing region subjected to methylation through RDR6-RdDM. (**A**) Sanger sequencing chromatograms at SNPs of *ALN*. WT (Cvi) plants cultivated at 22 ℃ or 10 ℃ were pollinated with WT (Col) pollen. RNA was extracted from endosperm of F1 seeds and subjected to RT-PCR followed by Sanger sequencing. Nucleotides at SNP sites are highlighted in red: 'C' originates from Cvi and 'T' originates from Col. Representative results are shown from experiments repeated at least five times

*Figure 2 continued on next page*

*Figure 2 continued*

for each genotype. (**B**) DNA methylation in the *ALN* 5' and 3' flanking regions of the endosperm of mature seeds. Blues lines show regions where DNA methylation was studied. Dashed red line (500 bp) shows the region analyzed for DNA methylation levels. All remaining methylation data shown in this figure only correspond to this 500 bp region. Red, blue and green vertical lines represent CG, CHG, and CHH methylation levels, respectively. (**C**) Histograms show percentage of DNA methylation in the 500 bp region. DNA extracted from endosperm of WT seeds and RdDM mutant seeds produced at 22 ˚C or 10 ˚C was analyzed by sodium bisulfite sequencing. (**D**) Cold-induced and seed-specific DNA methylation. Seeds produced at 22 ˚C or 10 ˚C were sown and the resulting seedlings cultivated at 22 ˚C for three weeks prior to DNA extraction and sodium bisulfite sequencing analysis (Whole seedling).

DOI: https://doi.org/10.7554/eLife.37434.006

The following source data and figure supplement are available for figure 2:

**Source data 1.** *Figure 2C*: Percentages of DNA methylation levels in the POGO region.

DOI: https://doi.org/10.7554/eLife.37434.008

**Figure supplement 1.** DNA methylation in the POGO region.

DOI: https://doi.org/10.7554/eLife.37434.007

This indicated that cold-induced CHH methylation in mature seeds takes place in gametes or in fertilization tissues or both.

Previous reports showed that AGO6 accumulates in gamete precursor cells (*McCue et al., 2015*). Furthermore, published microarray data show that *AGO6* expression is rather tissue-specific, being expressed in shoot apex, mature pollen and developing seeds (*Schmid et al., 2005*). In contrast, *RDR2*, *RDR6*, and *AGO4* expressions are more ubiquitous and higher relative to that of *AGO6* (*Schmid et al., 2005*).

We therefore hypothesized that cold-induced POGO region CHH methylation in mature seeds results from cold-induced stimulation of *AGO6* expression in gametes or in fertilization tissues or both. To test the hypothesis, we analyzed AGO6-GFP protein localization in developing seeds using an endogenous *AGO6* promoter reporter line (*pAGO6:AGO6-GFP*) (*McCue et al., 2015*). Consistent with microarray data, we observed AGO6-GFP fluorescence in seeds developing at 22 ˚C. Strikingly, cold (10 ˚C) markedly increased the AGO6-GFP fluorescence in the pre-fertilization ovule and in developing seeds relative to warm temperatures (22 ˚C) (*Figure 3A*). Consistent with microarray data, at 22 ˚C *AGO6* mRNA expression is low in pre-fertilization ovules relative to stamens (*Figure 3B*). At 10 ˚C, *AGO6* mRNA expression is induced in pre-fertilization ovules (*Figure 3B*). This suggests that methylation of the POGO region could be stimulated by cold in the female gamete prior to fertilization. Furthermore, we found that *AGO6* expression was higher in stamen and early developing seeds in plants cultivated at 10 ˚C relative to plants cultivated at 22 ˚C (*Figure 3B*). Expression levels of *AGO4* were also increased in seeds developing at 10 ˚C relative to those developing at 22 ˚C (*Figure 3B*). Expression levels of *RDR2* and *RDR6* were not significantly changed in response to cold (*Figure 3A and B*).

Taken together these results suggest that cold temperatures could stimulate *AGO6* expression and product accumulation in gametes and fertilization tissues. They support the hypothesis that cold stimulates RDR6-RdDM in male and female reproductive tissues or early developing seeds or both, thus leading to increased CHH methylation of the POGO region in mature seeds (*Figure 4—figure supplement 3*). In turn, increased methylation of *ALN* maternal allele would lead to loss of imprinting.

Furthermore, seedlings arising from seeds produced at 10 ˚C or at 22 ˚C had similar methylation levels in the POGO region (*Figure 2D*). This strongly suggests that after germination cold-induced CHH methylation is lost as the combined result of cell division and absence of *AGO6* expression in seedlings (*McCue et al., 2015*).

## Methylation changes in the POGO region correlate with changes in *ALN* expression and dormancy levels

We investigated whether changes in the POGO region CHH methylation are associated with changes in *ALN* endosperm expression levels. *ALN* endosperm expression upon seed imbibition was higher in WT seeds produced at 22 ˚C relative to seeds produced at 10 ˚C, as previously shown (*Figure 4A*) (*Piskurewicz et al., 2016*). In *nrpd1* mutant seeds produced at 10 ˚C, in which CHH methylation is increased by cold, *ALN* expression was suppressed similarly to WT (*Figure 4A*). In

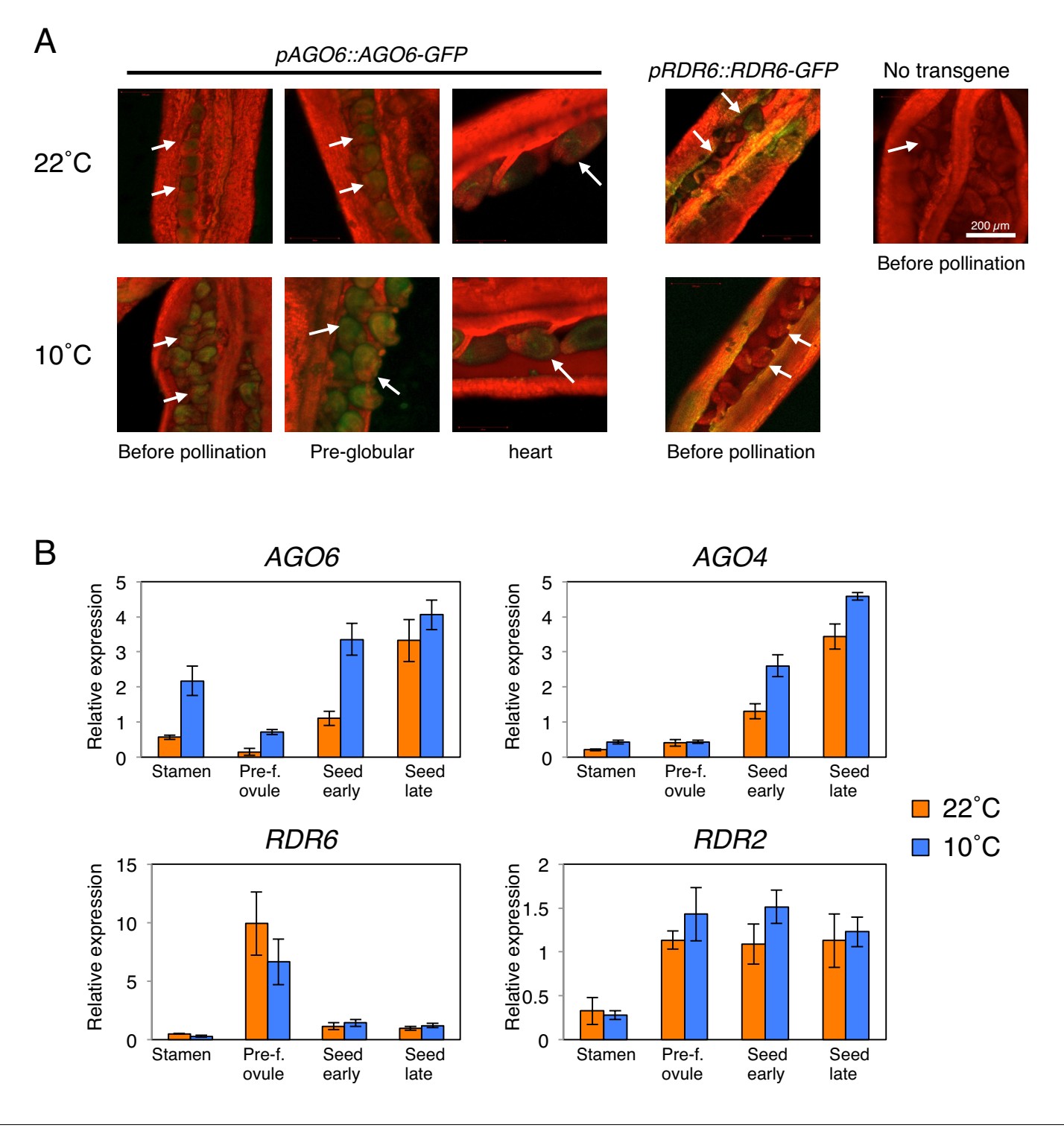

**Figure 3.** *AGO6* expression is tissue-specific and is stimulated by cold. (**A**) *pAGO6::AGO6-GFP* and *pRDR6::RDR6-GFP* transgenic lines were used to image AGO6-GFP and RDR6-GFP fluorescence, respectively, at various stages of ovule and seed development using confocal laser-scanning microscopy. Plants lacking the transgene are shown as a negative control (No transgene). Chlorophyll auto-fluorescence appears in red. Arrows indicate pre- and after-fertilization ovules. (**B**) qRT-PCR analysis of *AGO6*, *AGO4*, *RDR2*, and *RDR6* mRNA accumulation in plants cultivated at 22°C and 10°C. RNA was extracted from stamens (Stamen), pre-fertilization ovules (Pre-f. ovule), developing seeds at the heart stage (Seed early) and developing seeds at the mature green stage (Seed late). Expression levels were normalized to those of *PP2A*. Error bars indicate SD from three technical replicates. Experiments were repeated two times giving similar results.

*Figure 3 continued on next page*

*Figure 3 continued*

DOI: https://doi.org/10.7554/eLife.37434.009

The following source data is available for figure 3:

**Source data 1.** *Figure 3B*: qRT-PCR expression analysis of *AGO6, AGO4, RDR2,* and *RDR6.*

DOI: https://doi.org/10.7554/eLife.37434.010

contrast in *drm* mutants produced at 10 ˚C, *ALN* expression was comparable to that found in *drm* seeds produced at 22 ˚C.

To further investigate the role of DNA methylation to regulate *ALN* gene expression, we generated transgenic lines where the expression of the *GUS* reporter gene is controlled by sequences upstream the *ALN*'s TSS that either lack (p*ALN*-845) or contain (p*ALN*-1525) the highly methylated POGO region (*Figure 4B*). In three independent p*ALN*-1525 transgenic lines, harboring the POGO region, *GUS* expression was suppressed in seeds produced at 10 ˚C relative to that in seeds produced at 22 ˚C (*Figure 4B*). In contrast in three independent p*ALN*-845 lines, lacking the POGO region, *GUS* expression levels were similar between seeds produced at 22 ˚C and those produced at 10 ˚C (*Figure 4B*).

These results suggest that the POGO region indeed contains regulatory elements necessary to suppress *ALN* transcription in response to cold during seed development. They support the notion that cold-induced CHH methylation suppresses *ALN* expression upon seed imbibition.

*ALN* is a negative regulator of seed dormancy and *ALN* expression levels affect seed dormancy levels (*Piskurewicz et al., 2016*). We analyzed seed dormancy levels in various RdDM mutant seeds produced at 22˚C and 10˚C. WT and RdDM mutant seeds produced at 22 ˚C had no detectable dormancy after a few days of after-ripening (*Figure 4—figure supplement 1*). *ago4* and *rdr2* mutant seeds produced at 10 ˚C increased their dormancy similarly to that of WT seeds. However, cold-produced *drm, nrpe1, ago6,* and *rdr6* mutant seeds, unable to increase POGO region methylation, were less dormant compared to WT (*Figure 4C* and *Figure 4—figure supplement 1*).

These results establish a strong positive correlation between POGO region methylation and dormancy levels in seeds produced at 10 ˚C. However, we also observed that *nrpd1* seeds produced at 10 ˚C were less dormant than WT seeds even though they increased CHH methylation (*Figure 4C* and *Figure 4—figure supplement 1*). One possible explanation for this apparent discrepancy is if Pol IV promotes dormancy irrespective of temperature during seed development. To better reveal differences in dormancy levels of seeds produced at 22 ˚C, we used suboptimal germination conditions (*De Giorgi et al., 2015*). After one week of after-ripening, *nrpd1* seeds were able to germinate unlike WT and *drm* seeds, indeed suggesting Pol IV promotes seed dormancy irrespective of the temperature (*Figure 4—figure supplement 1*).

As expected, seeds produced at 10 ˚C from plants overexpressing *ALN* had low dormancy levels (*Figure 4C* and *Figure 4—figure supplement 1*). Furthermore, seeds produced at 10 ˚C from *aln drm1 drm2* and *aln drm1 drm2 cmt3* mutants had high dormancy levels similar to WT seeds (*Figure 4C* and *Figure 4—figure supplement 1*). These observations strongly suggest that high *ALN* expression in *drm* mutants contributes to their lower seed dormancy levels.

It is reported that RNAi targeted to specific loci can induce de novo DNA methylation (*Mette et al., 1999*; *Mette et al., 2000*; *Matzke et al., 2004*). To further study the role of POGO region methylation in cold-dependent *ALN* expression suppression and regulation of dormancy, we generated transgenic lines expressing an inverted repeat construct to be transcribed into dsRNA in order to generate siRNAs targeting the POGO region. In the T2 generation of transgenic plants, different dormancy levels were observed among different transgenic lines cultivated at 22 ˚C (*Figure 4D* and *Figure 4—figure supplement 2*). Among 22 transgenic lines, only four lines produced seeds with dormancy levels similar to WT seeds. The remaining 18 lines had higher dormancy levels compared to WT seeds. We selected four lines producing seeds with the highest dormancy levels and two lines producing seeds with dormancy levels similar to WT seeds. In the low dormancy lines endogenous POGO region CHH methylation and *ALN* expression levels were similar to those of WT plants (*Figure 4D* and *Figure 4—figure supplement 2*). In contrast, in the high dormancy lines endogenous POGO region CHH methylation levels were markedly increased relative to those in WT plants and *ALN* expression was suppressed (*Figure 4D* and *Figure 4—figure supplement 2*).

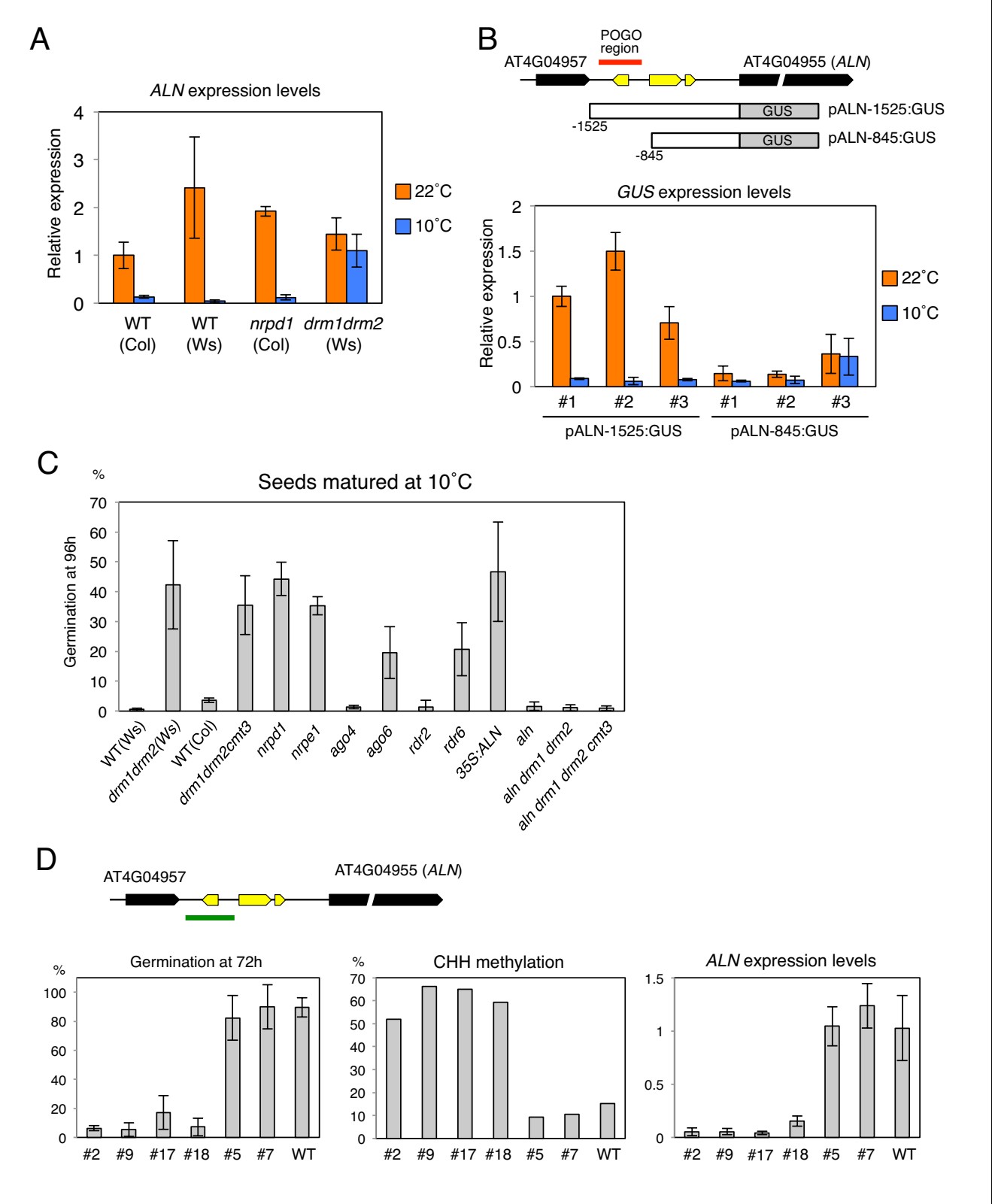

**Figure 4.** Changes in the POGO region methylation levels correlate with changes in *ALN* expression and dormancy levels. (A) qRT-PCR analysis of *ALN* mRNA accumulation in WT, *nrpd1* and *drm1drm2* endosperm. RNA was extracted from endosperm of seeds produced at 22℃ and 10℃ and imbibed for 48 hr. *ALN* expression levels were normalized to those of *PP2A*. Error bars indicate SD of results from three technical replicates. Experiments were repeated three times giving similar results. (B) Analysis of reporter expression driven by *ALN* upstream sequences. Two different lengths of *ALN*

*Figure 4 continued on next page*

*Figure 4 continued*

upstream sequences (open box) were fused to GUS gene (filled box) and transformed into WT plants (Ws background). RNA was extracted from endosperm of seeds produced at 22°C and 10°C and imbibed for 48 hr. *GUS* expression levels were normalized to those of *PP2A*. Analysis was performed on three independent transgenic lines. Error bars indicate SD of results from three technical replicates. (C) Histograms show percent germination in populations of seeds produced 10 °C. Germination was scored 96 hr after seed imbibition (three biological replicates, n = 80–120). *drm1drm2* is in Ws background; all other mutants are in Col-0 background. (D) Inducing DNA methylation of the *AthPOGON1*-containing region upstream of *ALN*. A DNA fragment covering *AthPOGON1* (indicated by a green line) was cloned into a RNAi vector to produce double stranded inverted repeats (Materials and methods) and transformed into WT plants (Ws background). Dormancy levels, CHH methylation levels of endogenous *ALN* upstream sequences (Materials and methods) and *ALN* expression levels were analyzed for independent transgenic lines as shown. For the analysis of dormancy levels, seeds were after-ripened for 10 days, and germination was scored 72 hr upon seed imbibition (three replicates, n = 100). RNA was extracted from endosperm 48 hr after imbibition. *ALN* expression levels were normalized to those of *PP2A*. Error bars indicate SD of results from three technical replicates.

DOI: https://doi.org/10.7554/eLife.37434.011

The following source data and figure supplements are available for figure 4:

**Source data 1.** *Figure 4A, 4B, 4C and 4D*: qRT-PCR expression analysis of *ALN*, GUS. Germination percentages of populations of seeds of different genotypes and produced at 10°C.

DOI: https://doi.org/10.7554/eLife.37434.015

**Figure supplement 1.** Germination percentages of populations of seeds of different genotypes and produced under different temperatures

DOI: https://doi.org/10.7554/eLife.37434.012

**Figure supplement 2.** Germination percentages and POGO region DNA methylation levels of different transgenic lines

DOI: https://doi.org/10.7554/eLife.37434.013

**Figure supplement 3.** Model for parental and environmental control of seed dormancy through epigenetic control of *ALN* expression.

DOI: https://doi.org/10.7554/eLife.37434.014

Altogether, these results support a model where cold-induced and RDR6-RdDM-dependent *ALN* promoter CHH methylation promotes seed dormancy by repressing *ALN* expression in seeds (*Figure 4—figure supplement 3*).

## Discussion

### *ALN* imprinting is implemented by non-canonical RdDM

Here we showed that *ALN* preferential maternal allele expression depends on DNA methylation of the paternal allele through non-canonical RdDM rather than MET1 as previously proposed for numerous MEGs (*Gehring, 2013*; *Rodrigues and Zilberman, 2015*). An association between RdDM and imprinted gene expression early upon fertilization has previously being described showing that some imprinted genes require canonical RdDM for paternal allele suppression; however, they also require MET1 activity (*Vu et al., 2013*).

The low abundance of CG sites in the *ALN* promoter suggests that CG methylation is not involved in suppressing *ALN* expression. In contrast, we showed that CHH methylation of POGO region located upstream of *ALN*'s TSS positively correlate with low *ALN* expression. This indicates that CHH methylation mediated by RdDM regulates *ALN* gene expression.

### Developmental origin of the imprinting marks

Our results indicate that *ALN* paternal allele CHH methylation is established in sperm cells during gametogenesis and maintained during seed development, which may be due to non-canonical RdDM specifically taking place in the male gamete lineage or fertilization tissues or both (*Figure 4—figure supplement 3*).

However, it remains to be understood how higher CHH methylation levels are maintained in the paternal allele relative to the maternal allele after fertilization (*Figure 1C*). Guiding siRNAs for RdDM are not expected to discriminate paternal from maternal alleles. Indeed, maternal *ALN* allele CHH methylation levels are higher in the endosperm relative to those observed in flower and seedling tissues though they remain, in the endosperm, lower relative to those of the paternal *ALN* allele (*Figure 1D and E*).

One possible explanation for this discrimination is that RdDM is more efficient on methylated DNA, thus maintaining higher paternal *ALN* allele methylation over time after fertilization. Consistent

with this hypothesis, Pol V is preferentially associated with methylated loci (*Wierzbicki et al., 2012*). It was shown that the SET domain proteins SUVH2 and SUVH9 bind to methylated DNA and facilitate the recruitment of Pol V to RdDM-targeted loci (*Liu et al., 2014*). Thus, after DNA replication, Pol V might be preferentially associated with hemi-methylated paternal chromosome, which would then facilitate DRM2-dependent CHH methylation of both strands.

Another possibility is that higher paternal allele CHH methylation levels at the time of fertilization serve as a primary imprint, which can generate a secondary imprint such as histone modification. This secondary imprint would then attract RdDM and thus maintain higher levels of CHH methylation in the paternal *ALN* allele. In plants, H3K27 trimethylation was shown to serve as a secondary imprint in some paternally expressed genes (*Gehring, 2013*; *Rodrigues and Zilberman, 2015*). In these cases, DNA hypomethylation of the maternal allele promotes H3K27 trimethylation, which suppresses maternal allele expression. This scenario could also operate in the case of *ALN*. Consistent with th possibility that H3K27 trimethylation could serve as a secondary imprint, a recent report showed that H3K27 trimethylation also takes place in hypermethylated paternal alleles (*Moreno-Romero et al., 2016*).

## Cold-dependent RdDM in mature seeds is reset after germination

Previous studies associated environmental cues, including cold temperatures, with changes in CHH methylation levels in Arabidopsis and different plant species (*Dubin et al., 2015*; *Secco et al., 2015*; *Wibowo et al., 2016*; *Secco et al., 2017*). In most cases changes are transient, which is consistent with the notion that CHH methylation can be a dynamic process contrarily to CG methylation, which is faithfully maintained by MET1 after each cell division (*Kawakatsu et al., 2017*).

Here we found that CHH methylation levels of the POGO region located upstream of *ALN*'s TSS are stimulated by cold temperatures during seed development, which represses *ALN* expression, thus promoting seed dormancy. Interestingly, increased cold- and RDR6-RdDM-dependent stimulation of CHH methylation is observed in seed tissues but not in flowers. This reflects the tissue-specific activity of RDR6-RdDM (*Figure 3A and B*). Cold-induced CHH methylation is lost in the seedling after germination (*Figure 2D*).

It has been reported that there are resetting mechanisms of epigenetic changes induced by environmental cues (*Choi et al., 2009*; *Crevillén et al., 2014*; *Iwasaki and Paszkowski, 2014*; *Crisp et al., 2016*). One notable example is *FLC*, encoding a repressor of flowering. *FLC* expression is suppressed by H3K27 trimethylation by prolonged cold (vernalization). H3K27 trimethylation-imposed suppression of *FLC* expression is maintained after the vernalization period and this epigenetic mark is then removed during late embryo development (*Berry and Dean, 2015*). This 'resetting' mechanism ensures that the plant's progeny can once again adapt appropriately their flowering time in a new environment.

Similarly, we propose that cold-induced CHH methylation of the POGO region located upstream of *ALN*'s TSS is a seed-specific mechanism allowing the seed to keep information of past cold temperatures in order to optimize seed germination timing. After germination, the information is erased in the embryo thus allowing optimal gene expression once again in the next generation.

## Materials and methods

### Plant material

Arabidopsis mutants, *met1-1* (*Kankel et al., 2003*), *nrpd1a-3* (*Herr et al., 2005*), *drm1-1 drm2-1* (*Cao and Jacobsen, 2002*), *nrpe1* (SALK_017795), *ago4* (SALK_027933), *ago6-2* (SALK_031553), *ago4-1 ago6-1* (*Zheng et al., 2007*), *rdr2* (SALK_206644), *rdr6-11* (*Peragine et al., 2004*), *rdr2-1 rdr6-15* (*Garcia-Ruiz et al., 2010*) were obtained from The European Arabidopsis Stock Centre (RRID:SCR_004576). *pAGO6::AGO6-GFP* (*McCue et al., 2015*) and *pRDR6::RDR6-GFP* (*McCue et al., 2012*) transgenic lines were kindly provided by Dr. Keith Slotkin. *35S :ALN* (*Watanabe et al., 2014*) construct was kindly provided by Dr. Atsushi Sakamoto.

### Plant growth conditions and germination assays

Plants were grown at 21–23°C, 16 hr/8 h day/night photoperiod, light intensity of 80 µE·m−2·s−1, humidity of 70%. For cold temperature treatment, plants were transferred to growth cabinets at

10°C after bolting. For germination assays, seeds were surface-sterilized (5% Bleach, 0.05% Tween) and sown on a Murashige and Skoog medium containing 0.8% (w/v) Bacto-Agar (Applichem). Between 80 and 150 seeds were examined with a Stemi 2000 (Zeiss) stereomicroscope and photographed with a high-resolution digital camera (Axiocam Zeiss) at different times of seed imbibition.

## Plasmid construction and plant transformation

For the *GUS* reporter gene construct, a 1525 bp fragment spanning −1525 to −1, and a 845 bp fragments spanning −845 to −1 relative to the TSS of *ALN* were amplified from WT Col-0 genome by PCR and cloned into *Bam*HI and *Sal*I sites flanking the *GUS* coding region of pBI101 plasmid. For the RNAi vector, the 601 bp flagment spanning −846 to −1446 bp relative to *ALN*'s TSS was similarly amplified by PCR and cloned into the pRNAi-GG plasmid following the protocol from *Yan et al. (2012)*. The *35S::ALN* plasmid was described previously (*Watanabe et al., 2014*). The resulting constructs were introduced into *Agrobacterium tumefaciens* GV3101 strains to transform wild-type Col-0 or Ws plants by the floral-dip method as previously described (*Clough and Bent, 1998*). Plants from T2 and T3 generation were used for analysis. Primers used in this study are listed in *Supplementary file 1*.

## DNA methylation analysis

Genomic DNA from embryo and endosperm was extracted as described previously (*Piskurewicz and Lopez-Molina, 2011*). Genomic DNA from flower and seedling was extracted with DNeasy Plant Mini Kit (Qiagen). Bisulfite DNA conversion was performed using EpiTect Bisulfite Kits (Qiagen). PCR was performed with primers listed in *Supplementary file 1*. Amplified fragments were cloned into the pGEM-T Easy Vector (Promega) and 10 to 12 independent clones for each sample were sequenced. Experiments were repeated at least twice. Sequencing data were analyzed with Kismeth (http://katahdin.mssm.edu/ kismeth).

## RNA extraction and RT-qPCR

Total RNA was extracted as described previously (*Piskurewicz and Lopez-Molina, 2011*). Total RNAs were treated with RQ1 RNase-Free DNase (Promega) and reverse-transcribed using ImpromII reverse transcriptase (Promega) and oligo(dT)15 primer (Promega). Quantitative RT–PCR was performed by using the ABI 7900HT fast real-time PCR system (Applied Biosystems) and Power SYBR Green PCR master mix (Applied Biosystems). Relative transcript levels were calculated using the comparative ΔCt method and normalized to the *PP2A* (*AT1G69960*) gene transcript levels. Primers used in this study are listed in *Supplementary file 1*.

## Confocal microscopy

The confocal fluorescent images were obtained by a Zeiss LSM 700 confocal microscope and ZEN microscope and imaging software package (Zeiss). GFP signals were taken with 488 nm laser and 520/35 nm band-pass filter. Chlorophyll autofluorescence signals were taken with 639 nm laser and 640 LP band-pass filter.

## Acknowledgments

We are especially grateful to Keith Slotkin and Olivier Voinnet for generously sharing seed material. We thank Atsushi Sakamoto for providing the *35S::ALN* plasmid. We thank Olivier Voinnet for helpful discussions.

We thank all members of the LLM laboratory for discussions. This work was supported by grants from the Swiss National Science Foundation and by the State of Geneva.

## Additional information

### Funding

| Funder | Grant reference number | Author |
|---|---|---|
| Swiss National Science Foundation | 31003A-152660/1 | Luis Lopez-Molina |
| Swiss National Science Foundation | 31003A-179472/1 | Luis Lopez-Molina |

The funders had no role in study design, data collection and interpretation, or the decision to submit the work for publication.

### Author contributions

Mayumi Iwasaki, Conceptualization, Data curation, Formal analysis, Supervision, Investigation, Writing—original draft, Writing—review and editing; Lena Hyvärinen, Urszula Piskurewicz, Data curation, Formal analysis; Luis Lopez-Molina, Conceptualization, Supervision, Funding acquisition, Project administration, Writing—review and editing

### Author ORCIDs

Mayumi Iwasaki (iD) http://orcid.org/0000-0001-6561-4247
Luis Lopez-Molina (iD) http://orcid.org/0000-0003-0463-1187

### Decision letter and Author response

Decision letter https://doi.org/10.7554/eLife.37434.028
Author response https://doi.org/10.7554/eLife.37434.029

## Additional files

### Supplementary files

• Supplementary file 1. Primers used in this study.
DOI: https://doi.org/10.7554/eLife.37434.016

• Transparent reporting form
DOI: https://doi.org/10.7554/eLife.37434.017

### Data availability

All data generated or analysed during this study are included in the manuscript and supporting files.

The following previously published datasets were used:

| Author(s) | Year | Dataset title | Dataset URL | Database and Identifier |
|---|---|---|---|---|
| Calarco JP, Borges F, Donoghue MT, Van Ex F, Jullien PE, Lopes T, Gardner R, Berger F, Feijó JA, Becker JD, Martienssen RA | 2012 | Reprogramming of DNA methylation in pollen guides epigenetic inheritance via small RNA. | https://www.ncbi.nlm.nih.gov/geo/query/acc.cgi?acc=GSE40501 | NCBI Gene Expression Omnibus, GSE40501 |
| Nishimura T, Zilberman D | 2012 | Active DNA demethylation in plant companion cells reinforces transposon methylation in gametes | https://www.ncbi.nlm.nih.gov/geo/query/acc.cgi?acc=GSE38935 | NCBI Gene Expression Omnibus, GSE38935 |
| Park K, Kim MY, Vickers M, Park J, Hyun Y, Okamoto T, Zilberman D, Fischer RL, Feng X, Choi Y, Scholten S | 2016 | DNA demethylation is initiated in the central cells of Arabidopsis and rice | https://www.ncbi.nlm.nih.gov/geo/query/acc.cgi?acc=GSE89789 | NCBI Gene Expression Omnibus, GSE89789 |
| Hsieh P, He S, Buttress T, Gao H | 2016 | DNA methylation inheritance across generations | https://www.ncbi.nlm.nih.gov/geo/query/acc. | NCBI Gene Expression Omnibus, |

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
