## [Decision Letter]

Thank you for submitting your article "Non-canonical RNA-directed DNA methylation participates in maternal and environmental control of seed dormancy" for consideration by *eLife*. Your article has been reviewed by four peer reviewers, including Daniel Zilberman as the Reviewing Editor and Reviewer #1, and the evaluation has been overseen by Detlef Weigel as the Senior Editor. The following individuals involved in review of your submission have agreed to reveal their identity: Steven Penfield (Reviewer #2); Xiaoqi Feng (Reviewer #3).

The reviewers have discussed the reviews with one another and the Reviewing Editor has drafted this decision to help you prepare a revised submission.

Summary:

The authors of this manuscript report that the non-canonical AGO6-RdDM pathway establishes imprinted expression of the dormancy regulator *ALN* in the endosperm. The authors show that (i) methylation of the paternal *ALN* allele depends on Pol V, DRM1/2, AGO4/6 and RDR2/6, (ii) that CHH methylation on *ALN* is present in sperm, (iii) that CHH methylation levels increase at lower temperatures, (iv) that AGO6 and AGO4 are expressed higher in seeds at lower temperatures, (v) that dormancy levels of seeds produced at lower temperature depend on the regulators of the non-canonical RdDM. Based on this data the authors propose that *ALN* is methylated by the AGO6-RdDM in sperm and that this allele-specific CHH methylation is maintained after fertilization. The authors discuss two scenarios that may explain how allele-specific RdDM could be maintained after fertilization. The report of environmentally induced RdDM regulating gene expression, imprinting and seed dormancy is potentially very interesting. There are, however, several points the authors should address. In general, the reviewers were convinced that temperature affected AGO6 expression and dormancy. The reviewers were not convinced that parent-of-origin effects on *ALN* were relevant, as it is not clear how *ALN* variation affects dormancy at low temperatures. The reviewers were also not convinced that methylation of *AthPOGON1* regulates *ALN* expression, or that the effects of the RdDM pathway on dormancy go through *ALN*.

Essential revisions:

The revisions requested below are meant to clearly demonstrate that DNA methylation-regulated imprinted expression of *ALN* is modulated by temperature and responsible for the reduced dormancy of RdDM mutants in the cold. Not all the suggested experiments may be required to do this, and there may be alternate approaches that the authors are welcome to employ.

1) A major weakness of this paper is that although the methylation patterns in WT and mutants are correlated to gene expression, there isn't an experiment showing that methylation of a particular sequence causes *ALN* expression changes. If the methylation covered a known regulatory region, or the transcriptional start site, this would be less of a concern. However, the causal relationship between the RdDM-induced DNA methylation and the imprinting of *ALN* was deduced from the DNA methylation at a small (300 bp) *AthPOGON1* TE that is about 1.2 Kb upstream of the *ALN* TSS.

This is of concern in part because the 5' prime region of *ALN* is generally TE rich, containing a repetitive region of 3 Kb that is only 400 bp upstream of *ALN* TSS; so is the 3' prime of *ALN* (a repetitive region about 5 kb including a TE overlapping *ALN* TTS). The available DNA methylome data from seed and pollen, and siRNA data from pollen and leaves, suggest that *AthPOGON1* does not behave differently from other repeats in *ALN*'s promoter: the whole 3 Kb repetitive region is hypomethylated in the maternal endosperm and hypermethylated in sperm. Therefore, only focusing on the methylation pattern at *AthPOGON1* is misleading (the authors have no evidence that this specific element matters), and it is necessary to extend the DNA methylation analyses to cover the 5' and 3' prime regions of *ALN*.

2) Extending the methylation analyses to the larger repetitive regions around *ALN* may also help to explain the conflicting results shown in Figure 1: Figure 1A shows imprinting of *ALN* does not require Pol IV, however, Figure 1B shows AGO4 and RDR2 have larger impacts on the endosperm methylation than the non-canonical components AGO6 and RDR6, respectively. Additionally, published *rdr2* mutant sperm data show that the hypermethylation at *ALN* promoter (and the 3' prime region) is entirely RDR2 dependent. Based on Figure 1B and published sperm data, it would be surprising if canonical RdDM does not contribute to *ALN* imprinting at normal conditions. Analysing a larger region and/or replicating the experiments would sharpen the data and potentially alter some of the conclusions regarding the roles of canonical vs. non-canonical RdDM.

3) After a more thorough investigation of methylation around *ALN*, the authors should consider targeting methylation to the most likely regulatory region. If methylation that is targeted to a specific part of the *ALN* promoter (or 3' region) in the mother plant, in a way that recapitulates the methylation phenotype at 10°C, shuts down maternal *ALN* expression and increases dormancy, this would constitute strong evidence for the authors' claims about environmental and epigenetic regulation of *ALN* expression and dormancy, even if RdDM may also regulate dormancy in other ways.

4) The authors should consider using genetics to demonstrate that (at least some of) the effects of RdDM on dormancy go through *ALN*, for example by making *aln;polV* double mutants.

5) The authors ignore the potential role of the DME demethylase in establishing imprinting. The hypomethylation patterns upstream of *ALN* in the maternal endosperm and vegetative cell are both dependent on the DME demethylase. The TTS-spanning repetitive region 3' prime of *ALN* is also hypomethylated in the same fashion by DME in the seed and pollen. These published data should be integrated into the authors' models.

6) The authors should show that maternal regulation of *ALN* is important for the observed phenotypes. The latter could be tested by comparing germination levels of seeds derived from the cross *aln* x wild type and *aln* x *nrpe1*. Based on previously published work of the authors, seeds from the first cross should have increased dormancy, which is expected to be lower in the cross using *nrpe1* as pollen donor that should cause *ALN* activation. The authors could also cross *aln* with *nrpe1*/+. If CHH methylation is established in sperm, in an *nrpe1* heterozygous mutant a population of early and late germinating seeds would be expected.

7) Statistical treatment should be improved. It is unclear on how many replicates all analyses are based on. If error bars are provided it is unclear whether they are based on biological or technical replicates.

8) The authors propose that cold-induced CHH methylation in mature seeds is the result of seed-specific DNA methylation. Nevertheless, in their model in Figure 4—figure supplement 2B they propose that at 10 degrees the non-canonical RdDM pathway is active in the female gametophyte, apparently contradicting their findings. This requires clarification.

---

## [Author Response]

Essential revisions:The revisions requested below are meant to clearly demonstrate that DNA methylation-regulated imprinted expression of ALN is modulated by temperature and responsible for the reduced dormancy of RdDM mutants in the cold. Not all the suggested experiments may be required to do this, and there may be alternate approaches that the authors are welcome to employ.1) A major weakness of this paper is that although the methylation patterns in WT and mutants are correlated to gene expression, there isn't an experiment showing that methylation of a particular sequence causes ALN expression changes. If the methylation covered a known regulatory region, or the transcriptional start site, this would be less of a concern. However, the causal relationship between the RdDM-induced DNA methylation and the imprinting of ALN was deduced from the DNA methylation at a small (300 bp) AthPOGON1 TE that is about 1.2 Kb upstream of the ALN TSS.

We apologize for not have been explicit enough: in the first version of the manuscript we analyzed in fact the methylation of a region spanning -1500 to -200 bp (about 1.3. kbp) relative to *ALN*’s TSS but we only showed the data for the 300 bp region covering *AthPOGON1* (this 300 bp region is referred in the revised version as “POGO region”).

In the revised version of the manuscript we provide new methylation data for downstream *ALN*’s TTS and, more importantly, two new experiments that independently test the model that methylation of the POGO region leads to suppression of *ALN* expression: 1) a promoter reporter experiment using 5’ *ALN* flanking sequences that either contain or lack the POGO region (Figure 4B) and 2) an experiment where we promote methylation of the POGO region by means of a transgene expressing an inverted repeat containing POGO region sequences (Figure 4D). The results of these experiments strongly support our hypothesis (see further below).

This is of concern in part because the 5' prime region of ALN is generally TE rich, containing a repetitive region of 3 Kb that is only 400 bp upstream of ALN TSS; so is the 3' prime of ALN (a repetitive region about 5 kb including a TE overlapping ALN TTS). The available DNA methylome data from seed and pollen, and siRNA data from pollen and leaves, suggest that AthPOGON1 does not behave differently from other repeats in ALN's promoter: the whole 3 Kb repetitive region is hypomethylated in the maternal endosperm and hypermethylated in sperm. Therefore, only focusing on the methylation pattern at AthPOGON1 is misleading (the authors have no evidence that this specific element matters), and it is necessary to extend the DNA methylation analyses to cover the 5' and 3' prime regions of ALN.

We respectfully only partially agree with you. The “repetitive region of 3Kb” you refer to located 400 bp upstream of *ALN*’sTSS (delineated with a dashed line in Author response image 1) does not really consist of repetitive elements disposed over 3 Kb.

It contains a gene (AT4G04957) flanked on its 5’ side by a long and short TE (AT4TE11825 and AT4TE11820) and on its 3’ side by a few interspersed TEs that include *AthPOGON1* (AT4TE11815). Publicly available data suggest that only the region covering the gene At4G04957 is hypermethylated in all contexts. We have reproduced these data in Author response image 2.

**Author response image 2. respfig2:** 

Publicly available data suggest that:

1) In flowers (Lister et al., 2008) and seedlings (Stroud et al., 2013) the At4G04957 region is hypermethylated.

2) In seedlings CHH methylation present in the At4G04957 hypermethylated region is similarly reduced in *nrpd1* and *nrpe1* mutants, indicating that it is targeted by canonical RdDM.

3) In mature seeds (Kawakatsu et al., 2017), overall CHH methylation levels tend to increase in both the 3’ and 5’ sequences of *ALN*. This includes the POGO region and the region 5’ of AT4G04957 (covering part of AT4TE11825) but excludes the TEs (AT4TE11800, AT4TE805 and AT4TE810) located between *AthPOGON1* (AT4TE11815) and the *ALN’s* TSS, which are not markedly methylated.

Concerning our data about methylation in the 5’ side of *ALN,* in the first version of the manuscript we were showing in Figure 1 only methylation data in the POGO region (indicated by a red line in Author response image 2) but we had sequenced a region going from the 3’ end of At4G04957 to about 200 bp upstream of the *ALN* TSS, i.e. methylation data corresponding to 1.3 Kbp on the 5’ side of *ALN* (indicated by a blue line in Author response image 2). These methylation data are now shown in their entirety in the revised Figure 1. Our data are consistent with publicly available data, i.e. they show that in the 3’ flanking region of At4G04957 there is no hypermethylation as observed in the region covering At4G04957. Rather they show that only the POGO region that we depicted in the first version of the manuscript contains a substantial amount of methylation. The other TEs (AT4TE11800, AT4TE805 and AT4TE810) in the region going from 400bp upstream of the *ALN* TSS to 5’ of *AthPOGON1* (AT4TE11815) are not substantially methylated, consistent with publicly available data. We fully agree with you concerning the 3’ *ALN* region: it contains TEs and publicly available data indeed show that it is hypermethylated in all contexts in seedlings, flowers and seeds (also reproduced above). In the revised manuscript, we have performed sodium bisulfite over a 1.7 kbp interval in this region. Our data indeed confirm it is hypermethylated (the data are shown above and also shown in new Figure1). Importantly, our data show that in *nrpd1* mutants CHH methylation is mainly lost in the 3’ *ALN* region whereas it is retained in the POGO region located 5’ of *ALN* (new Figure 1).

One question you are raising is that of the importance of the hypermethylation 5’ of *ALN* (i.e., that found in the region covering At4G04957) and 3’ of *ALN* to suppress *ALN* gene expression. Firstly, it should be pointed out that publicly available data show that *ALN* expression is not suppressed in seedlings nor in flowers. Indeed *ALN* expression is comparable to that of *ALN’s* neighboring genes (AT4G04940, AT4G04960, AT4G04970). Thus *ALN* is an imprinted gene unlike many imprinted genes described by the imprinting “standard model” whereby DME activates the maternal allele expression in the central cell in a gene whose expression is otherwise shutdown by methylation throughout the plant’s life cycle. Secondly, publicly available data (see above)show that the *ALN*’s 5’ and 3’ hypermethylated regions substantially lose their methylation in *met1* or *drm1drm2cmt3 (ddc*) mutants (flower) and yet this is not associated with a substantial increase in *ALN* expression in flower (Lister et al., 2008) or seedling (Stroud et al., 2012) (see Author response image 3).

**Author response image 3. respfig3:** 

Thirdly, we showed that *ALN* imprinting is preserved in *met1* and *nrpd1* indicating that these hypermethylated regions play no substantial role to suppress the *ALN* paternal allele. Thus, there is no evidence suggesting that *ALN*’s 5’ and 3’ hypermethylated regions play a substantial role in suppressing *ALN* expression.

Publicly available data show that CHH methylation increases in mature seeds in a region upstream of At4G04957 covering part of AT4TE11825. We cannot rule out that this region, and indeed additional sequences in chromosome 4, is also targeted by non-canonical RdDM, in addition to the POGO region, and that it could also participate to suppress *ALN* expression. However, the data presented in the revised manuscript show that at least the POGO region and its methylation plays a significant role to repress *ALN* expression and promote dormancy.

2) Extending the methylation analyses to the larger repetitive regions around ALN may also help to explain the conflicting results shown in Figure 1: Figure 1A shows imprinting of ALN does not require Pol IV, however, Figure 1B shows AGO4 and RDR2 have larger impacts on the endosperm methylation than the non-canonical components AGO6 and RDR6, respectively. Additionally, published rdr2 mutant sperm data show that the hypermethylation at ALN promoter (and the 3' prime region) is entirely RDR2 dependent. Based on Figure 1B and published sperm data, it would be surprising if canonical RdDM does not contribute to ALN imprinting at normal conditions. Analysing a larger region and/or replicating the experiments would sharpen the data and potentially alter some of the conclusions regarding the roles of canonical vs. non-canonical RdDM.

We repeated several times the experiment in Figure 1A and we stand behind the conclusion that Pol IV is not necessary for *ALN* imprinting. Canonical RdDM is defined by the pathway functioning through Pol IV-RDR2-DCL3. So the observation that Pol IV is not necessary for imprinting implies that canonical RdDM is not necessary for imprinting. Given that *ALN* imprinting requires *DRMs* rather than MET1, it follows that *ALN* imprinting requires non-canonical RdDM. AGO4 and AGO6 act redundantly in many loci (Zheng et al., 2007). In addition, RDR2 and RDR6 can also function redundantly for antisense RNA synthesis (Matsui et al., 2017). We therefore do not see a conflict between Figure 1A and 1B: that imprinting does not require Pol IV does not exclude that AGO4 and RDR2 could have higher impact on methylation than AGO6 and RDR6, respectively. We now provide new data showing that *ALN* imprinting is lost in *ago4ago6* and *rdr2rdr6* (new Figure 1A) mutants, consistent with the low methylation in the POGO region (Figure 1B). We propose that the non-canonical form of RdDM responsible for *ALN* imprinting involves the components AGO4, AGO6, RDR2 and RDR6.

On the other hand, we propose that the RDR6-RdDM pathway (AGO6, RDR6) is specifically involved in cold-dependent suppression of *ALN* expression by promoting methylation in the POGO region.

3) After a more thorough investigation of methylation around ALN, the authors should consider targeting methylation to the most likely regulatory region. If methylation that is targeted to a specific part of the ALN promoter (or 3' region) in the mother plant, in a way that recapitulates the methylation phenotype at 10°C, shuts down maternal ALN expression and increases dormancy, this would constitute strong evidence for the authors' claims about environmental and epigenetic regulation of ALN expression and dormancy, even if RdDM may also regulate dormancy in other ways.

We studied the role of the *AthPOGON1*-containing region in cold-dependent *ALN* expression suppression and regulation of dormancy in two ways. First, we generated transgenic lines where the expression of the *GUS* reporter gene is controlled by the 5’ *ALN* flanking region that either lacks (p*ALN*-845) or contains (p*ALN*-1525) the highly methylated and *AthPOGON1*-containing region. In independent lines, GUS expression was suppressed in seeds produced at 10˚C in p*ALN*-1525 lines but not in p*ALN*-845 lines (New Figure 4B).

Second, we transformed WT plants with a transgene carrying an inverted repeat in order to generate siRNAs targeting the POGO region. Different lines were cultivated at 22˚C and different lines produced seeds with different levels of dormancy. Lines producing seeds with low dormancy levels had CHH methylation levels in thePOGO region and *ALN* expression levels similar to those of WT plants (New Figure 4D). In contrast, in lines producing seeds with high dormancy levels CHH methylation levels were markedly increased relative to those in WT plants and *ALN* expression was suppressed (New Figure 4D). There results support the model that the POGO region is important to suppress *ALN* expression and promote dormancy.

4) The authors should consider using genetics to demonstrate that (at least some of) the effects of RdDM on dormancy go through ALN, for example by making aln;polV double mutants.

We generated *aln/drm1/drm2* and *aln/drm1/drm2*/cmt3 mutants. Seeds produced at 10˚C from *aln/drm1/drm2* and *aln/drm1/drm2*/cmt3 mutants had high dormancy levels similar to WT seeds (New Figure 4C). These observations suggest that high *ALN* expression in *drm* mutants contributes to their lower seed dormancy levels.

5) The authors ignore the potential role of the DME demethylase in establishing imprinting. The hypomethylation patterns upstream of ALN in the maternal endosperm and vegetative cell are both dependent on the DME demethylase. The TTS-spanning repetitive region 3' prime of ALN is also hypomethylated in the same fashion by DME in the seed and pollen. These published data should be integrated into the authors' models.

Our imprinting model points to the role played by the 5’ flanking region spanning about 300bp containing the *AthPOGON1* TE element (referred as “POGO region” in the text and delineated by a red bar in Figure 1B and below). According to data published by Park et al., 2016, the POGO region has low methylation levels in the central cell independently of DME. These data are now shown in Figure 1F.

You are therefore probably referring to the hypermethylated region covering At4G04957 and indicated with a black asterisk in Author response image 4.

**Author response image 4. respfig4:** 

This region and the TTS-spanning repetitive region in the 3’ part of *ALN* are hypermethylated in seedlings (see also above). In *met1* mutant seedlings these regions become hypomethylated. Yet, *ALN* expression is similar in both WT and *met1* seedlings (see above). In the “standard model” describing the establishment of imprinting through the activity of DME in the central cell, imprinting genes are normally silenced by DNA methylation in seedlings (through CG methylation imposed by MET1). Thus, imprinting of *ALN* escapes the “standard model” in at least three counts: 1) It is a gene that is expressed seedlings; (2) Its expression in seedlings is not affected in *met1* mutants; (3) *ALN* imprinting persists in hybrid seeds using *met1* as a pollen donor (Figure 1A). Therefore we see no particular reason to include DME in the model.

6) The authors should show that maternal regulation of ALN is important for the observed phenotypes. The latter could be tested by comparing germination levels of seeds derived from the cross aln x wild type and aln x nrpe1. Based on previously published work of the authors, seeds from the first cross should have increased dormancy, which is expected to be lower in the cross using nrpe1 as pollen donor that should cause ALN activation. The authors could also cross aln with nrpe1/+. If CHH methylation is established in sperm, in an nrpe1 heterozygous mutant a population of early and late germinating seeds would be expected.

Indeed, we previously showed that seed dormancy levels can be regulated by maternal *ALN* allele expression notably by showing that *aln* x WT hybrid F1 seeds obtained after pollinating *aln* plants with WT pollen have higher dormancy than WT seeds. In the revised manuscript we pollinated *aln* mutant plants with WT and *nrpe1* pollen. Dormancy levels in *aln* x *nrpe1* hybrid F1 seeds were lower relative to *aln* x *WT* hybrid seeds, further indicating that maternal regulation of seed dormancy requires suppression of *ALN* paternal allele by RdDM (Figure 1—figure supplement 1C). We did not cross *aln* with *nrpe1*/+. Our data suggest paternal CHH methylation is established in male germ linage, however we still do not know whether it happens before or after meiosis. If CHH methylation is established before meiosis, we would not see the difference between two population (*nrpe1*/+ and +/+). In this study, it is not the main purpose to clarify exactly when CHH methylation is established.

7) Statistical treatment should be improved. It is unclear on how many replicates all analyses are based on. If error bars are provided it is unclear whether they are based on biological or technical replicates.

We apologize for the omission to mention the statistical-related information. This information now appears in the figure legends and in the Materials and methods section.

8) The authors propose that cold-induced CHH methylation in mature seeds is the result of seed-specific DNA methylation. Nevertheless, in their model in Figure 4—figure supplement 2B they propose that at 10 degrees the non-canonical RdDM pathway is active in the female gametophyte, apparently contradicting their findings. This requires clarification.

We apologize for the lack of precision. We sought to identify at which developmental stage the cold-induced non-canonical RdDM in the POGO region takes place (thereafter “cold-induced methylation”). First, we could not detect cold-induced methylation in whole flower tissues. This suggests that the cold-induced methylation that we detect in endosperm and embryos is only taking place in gamete precursor cells (thus evading detection when analyzing methylation in DNA extracted from whole flower tissues) or only in fertilization tissues or both.

Consistent with this notion, publicly available data together with our AGO6-GFP accumulation data (Figure 3A) as well as our *AGO6* mRNA expression data (Figure 3B) show that at 10˚C before fertilization, *AGO6* expression and AGO6 accumulation is mainly detected in subparts of the flower: stamens and ovules. After fertilization, *AGO6* expression and AGO6 accumulation is localized in developing seeds although our data do not specify exactly where in developing seeds is the AGO6-GFP accumulation and *AGO6* expression to be found.

Given that *AGO6* is genetically required for cold-induced methylation in the endosperm and embryo, we conclude that cold-induced methylation results from cold-dependent activation of non-canonical RDR6-RdDM in gametes or fertilization tissues or both (Figure 4—figure supplement 3).

We have rewritten the part of the text (main text and model legend) describing these points to hopefully make it clearer (subsection “Cold-dependent and tissue-specific stimulation of AGO6 expression likely drives cold-induced methylation of the POGO region in seeds”, last paragraph).